# Reliability of the Polish Version of the Kinesiophobia Causes Scale (KCS) Questionnaire in Assessing the Level of Fear of Movement Among People Suffering from Chronic Nonspecific Low Back Pain

**DOI:** 10.3390/diagnostics15141746

**Published:** 2025-07-09

**Authors:** Edward Saulicz, Andrzej Knapik, Aleksandra Saulicz, Damian Sikora, Mariola Saulicz

**Affiliations:** 1Institute of Physiotherapy and Health Sciences, Jerzy Kukuczka Academy of Physical Education, 40-065 Katowice, Poland; 2Department of Adapted Physical Activity and Sport, School of Health Sciences in Katowice, Medical University of Silesia in Katowice, 40-751 Katowice, Poland; 3School of Public Health and Social Work, Queensland University of Technology (QUT), Kelvin Grove, QLD 4059, Australia

**Keywords:** kinesiophobia, chronic lower back pain, questionnaire, reliability

## Abstract

**Background:** The phenomenon of fear of movement is called kinesiophobia. Kinesiophobia is a significant factor that complicates the treatment process. Fear of movement and physical activity is a risk factor for the transformation of acute pain into chronic pain. Therefore, the assessment of the level of kinesiophobia is a prognostic factor for disability and mental stress, thus having a significant impact on the quality of life of people with lower back pain. One of the psychometric diagnostic tools for assessing the level of kinesiophobia is the Kinesiophobia Causes Scale (KCS). The aim of the study was to assess the reliability of the KCS test used in people suffering from chronic nonspecific lower back pain (nsLBP). **Methods:** The study included a group of 112 people suffering from chronic nsLBP. The subjects completed the same Polish version of the KCS questionnaire 4 weeks apart. **Results:** Good internal consistency was recorded for both domains—the biological and psychological one—as well as the general KCS index (Cronbach’s alpha index α from 0.8 to 0.9). Reliability was excellent for both domains (95% CI of ICC_3.1_ biological domain: 0.86–0.93 and for psychological domain: 0.92–0.96) and for the total score of the Kinesiophobia Causes Scale (95% CI of ICC_3.1_: 0.91–0.93). **Conclusions:** These results indicate very good measurement reliability of the Polish version of the KCS questionnaire among people suffering from chronic nsLBP.

## 1. Introduction

Due to its common occurrence, lower back pain (LBP) is considered the most common health problem worldwide, next to the common cold [1,2]. It is estimated that 70–85% of the world’s population has experienced LBP at least once in their life [3,4,5,6,7]. Most often, these symptoms are temporary; in 80–90% of cases, symptoms disappear after 12 weeks regardless of the applied treatment method [8,9,10]. However, a significant percentage of people, up to 75%, experience the recurrence of LBP [11,12], and in almost 30% of cases, the symptoms become chronic [13]. It is quite rare to identify a specific cause of LBP. The most prevalent type is mechanical nsLBP [14]. The scale of nsLBP occurrence makes it not only a medical problem burdening healthcare systems but also a socio-economic problem through lower work efficiency or sick leave [5,15,16,17,18,19,20,21,22]. Reducing the risk of occurrence and shortening the recovery time when nsLBP occurs could bring significant benefits both at the individual and societal level.

Fears of the occurrence or intensification of pain limit physical activity (PA) and cause fears of undertaking it. Fear of pain can change movement habits towards the development of so-called avoidance movement patterns. Avoidance movement patterns are often a source of further overloads and the development of new nociceptive foci in other parts of the body, taking the form of a vicious circle according to the Vlaeyen–Linton model [23,24,25,26,27,28]. Paradoxically, the strategy of avoiding PA is considered by many to be an appropriate and desirable reaction that has a positive impact on their health. Studies show that this may apply to over 70% of people suffering from chronic pain [29,30,31,32]. The avoidance of PA by patients with nsLBP is often phobic in nature. In psychology and behavioral therapy, phobias are considered acquired and subject to therapy through confrontational techniques that erase the previously encoded fear reaction [33]. Patient education explaining the processes of pain development and factors that may intensify it is a good way to support the treatment process. It also has a beneficial effect on reducing the fear of movement and the treatment process of nsLBP [34,35,36,37,38].

The irrational fear of movement resulting from the fear of pain is called kinesiophobia [39]. Kinesiophobia plays a significant role in the process of transforming acute pain into chronic pain, and a number of studies have shown that it is an important prognostic factor of disability and psychological stress, significantly affecting the quality of life [40,41,42,43]. The Tampa Scale of Kinesiophobia—TSK [32,44,45]—or its shortened version —TSK-11 [46]—is most often used to estimate its level. The importance of kinesiophobia has led to the development of a number of other tools, including the Fear and Avoidance Belief Questionnaire (FABQ), Fear and Pain Avoidance Scale (FAPS), Fear and Pain Avoidance Questionnaire in Athletes (AFAQ), NeckPix, and the Fear and Pain Questionnaire (FPQ) [47,48,49,50,51]. The common denominator of these diagnostic tools is the unequivocal connection of the fear of movement with the occurrence and intensity of pain. In our opinion, this simple mechanism of association, involving stimulus–reaction–imagination–behavior, does not fully exhaust either the causes or characteristics of this phenomenon. In relation to PA, anxiety symptoms typical of phobias occur very rarely; therefore, its significant limitation should be classified as avoidance behaviors. The cause of kinesiophobia is not only the fear of pain but also the fear of the consequences of PA in the form of experiencing physical and mental discomfort (muscle pain after exercise, feeling tired, exhausted, ridiculed due to lack of fitness, negative perception of forms of activity by the near and distant environment). This broader view of kinesiophobia is consistent with the biopsychosocial model of health perception and its disorders. This assumption was the basis for the development of the Kinesiophobia Causes Scale (KCS) questionnaire [52]. Despite its necessary reductionism, the KCS quite comprehensively assesses potential biological and psychological barriers related to PA and their severity. Since it has not been used so far in the diagnosis of kinesiophobia in people suffering from LBP, the aim of this study is to assess the validity and reliability of the KCS in diagnosing the level of kinesiophobia in people suffering from LBP incidents. The initial hypothesis was that the KCS questionnaire is characterized by good internal consistency. The hypothesis was also accepted about the high reliability of the Polish version of the KCS questionnaire among people with chronic nsLBP.

## 2. Materials and Methods

### 2.1. Ethical Considerations

This study is part of a broad research problem related to the assessment of environmental sources of health hazards, the plan of which was accepted by the local Bioethics Committee for Scientific Research operating at the Jerzy Kukuczka Academy of Physical Education in Katowice (no. 3/2022). Each person taking part in the study was required to sign a statement of informed consent to participate in the study in accordance with the guidelines of the Declaration of Helsinki developed by the World Medical Association.

### 2.2. Study Design and Subjects

The study was conducted in the southern part of Poland (provinces: Silesia, Lesser Poland, and Opole). The inclusion criteria for the study were freedom of choice, age over 18 years, and no diseases or dysfunctions requiring treatment other than LBP. The exclusion criterion was the current occurrence of nsLBP requiring therapy.

This was an observational, cross-sectional study conducted in 2022–2023. The study used a diagnostic survey method—a paper-and-pencil questionnaire. The metric part collected data on gender, age, confirmation or denial of the fact of having had an LBP incident, the number of incidents, the time that passed since the last incident, and its duration. On the horizontal VAS scale, the subjects marked the degree of pain intensity during the last pain incident. The Polish version of the modified Oswestry questionnaire was used to examine the limitations related to LBP [53].

The level of kinesiophobia in people suffering from nsLBP was assessed twice using the KCS questionnaire [52,54], which allowed for determining the level of barriers to physical activity in two domains, namely the biological and psychological domains. Each of these domains (subscales) contains four factors, and the scoring for the responses on a scale from 0 to 100 is intended to determine the level of intensity of barriers to activity. The biological domain is the average of the following factors: morphological, individual need for stimulation, energetic substrates, and power of biological drives. The psychological domain is the average of points from the following factors: self-acceptance, self-assessment of motor predisposition, state of mind, and susceptibility to social influence, while the total score of the Kinesiophobia Causes Scale is half of the sum of points from both domains—the biological and psychological ones [52,54,55].

All research material was subjected to individual selection stages (Figure 1).

In the last stage, every 4th person (25% of people with LBP), i.e., 122 people, were also given an envelope containing the second KCS questionnaire and an addressed envelope with a stamp attached, with a request to open the envelope after four weeks of completing the questionnaire and return it to the address written on the envelope. Since there are no clear recommendations regarding the time interval that should be used in studies assessing the repeatability of results in survey studies, a time interval of 1 month was arbitrarily adopted. The KCS questionnaires were marked with reference numbers that allowed for the later pairing of questionnaires from the first and second study.

Finally, in accordance with the aim of the study, questionnaires were obtained from 112 people who completed them twice, including 90 women (80.4%) and 22 men (19.6%). The age of the respondents ranged from 22 to 84 (49.8 ± 14.9) years.

### 2.3. Statistical Analysis

The calculations were performed using Statistica version 13.3, SPSS version 24.0 (StatSoft Inc., Tulsa, OK, USA), and MS Excel from Microsoft Office 2016 (Microsoft Corporation, Redmond, WA, USA). Descriptive statistics of the studied variables were performed. The interclass correlation coefficient (ICC) [56] was used to calculate reliability. The scale proposed by Portney and Watkins [57] was used to interpret the results. The reliability of the results was assumed to be high at ICC = 0.75–1.0; moderate at ICC = 0.50–0.75; and low at ICC < 0.50. Additionally, during reliability calculations, the standard error of measurement (SEM) was analyzed according to the formula SEM = SD × √1 ICC and 95% confidence intervals. The internal consistency of the questionnaires was assessed using Cronbach’s alpha, assuming a satisfactory value of α > 0.7 [58]. Using Slovin’s formula, the minimum sample size was also calculated, which given the original population size of 2045, was 145 people. A total of 487 people met the chronic LBP criteria, thus meeting the minimum sample size requirements.

## 3. Results

The majority of the study participants indicated minimal-to-moderate disability due to LBP (range of ± 1 SD Oswestry 8.5–35.5 points), and the pain intensity during the last LBP incident was estimated from 4.56 to 6.7 (mean ± 1 SD).

The internal consistency of the KCS was examined. The analysis of the α coefficient value (Table 1) indicates good internal consistency of both domains (95% CI 0.7–0.8 for the biological domain and 0.8–0.8 for the psychological domain) and approaching excellent for the entire KCS questionnaire (95% CI 0.8–0.9).

The data in Table 2 show that people with nonspecific LBP are characterized by a high degree of reliability for the total score of the Kinesiophobia Causes Scale, which was 0.93 (95% CI 0.91–0.95), 0.9 (95% CI 0.86–0.93) for the biological domain, and 0.94 (95% CI 0.92–0.96) for the psychological domain. The SEM value for both domains and the general index of kinesiophobia ranged from 0.01 to 0.2. All factors constituting both domains were also characterized by high reliability, and the ICC_3.1_ coefficient for the components of the biological domain ranged from 0.86 to 0.93; for the psychological domain, it ranged from 0.91 to 0.93. The SEM values for the factors constituting the biological domain ranged from 0.04 to 0.3, and for the factors constituting the psychological domain, they ranged from 0.01 to 0.4.

The results concerning repeatability in individual questions of the KCS questionnaire are presented in Appendix A in Appendix A. Out of 24 detailed questions (questions 8 and 13 contain sub-items a, b, and c), only the results from questions 9 and 13a indicate moderate reliability. In the case of the remaining 22 detailed questions, the ICC value indicates high reliability.

## 4. Discussion

The presented study results indicate high reliability and internal consistency of the KCS, which justifies the use of this questionnaire in accordance with the biopsychosocial approach to therapy. In the biopsychosocial approach to the therapy of people with chronic symptoms of LBP, the ability to function, disability, or health are measured not only by functional and structural disorders. Equally important is the assessment of ability in relation to self-sufficient activities and locomotion, as well as participation manifested by involvement in the social and professional environment [59]. In this approach, fear of pain will be a phenomenon that significantly hinders the process of recovery or normal functioning despite permanent functional deficits. The negative emotion of fear results in excessive sensitivity to any threats that may potentially result in pain. This will be manifested by the limitation of everyday natural physical activity resulting in the atrophy of skeletal muscles and a decrease in the efficiency of the cardiovascular system, which consequently intensifies pain reactions and leads to disability. Pain and disability, in turn, lead to emotional disorders, which in turn deepen avoidance behaviors and increase the fear of pain [60]. In this situation, exercise therapy aims to implement and convince individuals to participate in regular physical activity and to strengthen competences in the selection of means and implementation of regenerative activities [61]. In the biopsychosocial model, in which health problems are considered on three levels, namely the biological one, the psychological one, and the social one, an important element ensuring adequate treatment and an optimal model of healthcare is the identification of factors determining the disease. Health not only involves good physical condition but also good mental health and quality of life. The “bio-psycho-social” medical model proposes that diseases should be considered at both the biological, psychological, and social levels in order to identify factors determining the disease at these levels, provide reasonable treatment, and develop a model of healthcare. [62].

Kinesiophobia is not only a common fear of physical activity but also a belief that it is easy to hurt yourself during exercise [39,45]. It is one of the potential psychological factors leading to a deterioration in patients’ physical fitness and is also closely related to mental health disorders that seriously affect the results of rehabilitation and the quality of life of patients. Avoidance behaviors caused by fear of pain and the resulting negative effects must be changed [60]. In this process, the patient should be made aware that pain, especially in its chronic form, is not a manifestation of tissue damage, but it can often occur without a clear cause. In the process of learning to cope with pain, patient education is a key element [45,59,63,64]. The prerequisite for starting the process of educating the patient towards “taming” the fear of pain, overcoming it, and coping with the pain itself, mainly through gradually implemented adaptation to physical activity, is a well-developed process of diagnosing the level of fear of movement. One of the psychometric tools useful in this process is the KCS questionnaire [52]. This questionnaire equally identifies biological and psychological factors underlying kinesiophobic behaviors. It also takes into account the assessment of internal conditions (the patient and their predispositions and abilities) and external conditions (influences of the surroundings and social surroundings). In this context, the KCS questionnaire is not an alternative diagnostic tool to those currently used. Its purpose is not to assess the level of kinesiophobia itself. In other words, it is not a diagnostic tool that only tells what a given situation is like. The design of the questionnaire allows for establishing an individual patient profile, indicating the conditions (causes) of fear of undertaking physical activity. It therefore allows for a highly personalized process of therapeutic education of the patient aimed at optimally reducing their fear of movement. The high parity of the biological domain in this situation encourages the search for appropriate means in the form of selecting appropriate exercises and types of physical activity that allow, for example, patients to increase the level of fitness or coordination motor skills. The high level of the psychological domain indicates the need for greater educational impact in the process of functional therapy for patients with chronic LBP.

The study demonstrated the correct construction of the KCS questionnaire, as evidenced by its good internal consistency for both domains, namely the biological and psychological ones (α = 0.8), as well as for the questionnaire as a whole (α = 0.9).

Researchers do not agree on the period of time that should elapse between the first and second test in order to determine the reliability of psychometric tools. Kleine [65] recommended that this period should be at least 3 months. The aim is to eliminate memorizing the initial answers. The practice of conducting research indicates that there may be many factors that require shortening this period, and a number of repeated tests are conducted after two weeks [66,67]. These factors may include, for example, the possibility of losing contact with the subjects or a change in the background of the research, which could affect the results. In this study, the authors took these factors into account. It was decided that the “golden mean” would be a period of one month.

The test–retest repeatability of the KCS questionnaire among people with chronic nsLBP showed high reliability. This applies to both the biological domain (ICC_3.1_ = 0.90), the psychological domain (ICC_3.1_ = 0.94), and the total score of the Kinesiophobia Causes Scale (ICC_3.1_ = 0.93). All factors constituting both the biological and psychological domains obtained a high reliability of results (ICC_3.1_ ranging from 0.86 to 0.93). A similar high reliability of results applies to detailed questions constituting the KCS questionnaire, comprising 24 detailed questions (questions 8 and 13 appear in three versions) (Appendix A in Appendix A). Only in the case of two questions were the values of the obtained interclass correlation coefficient ICC indicating moderate reliability (ICC_3.1_ = 0.70 and 0.72). This concerned questions 9 and 13a. The first question concerns fatigue after work and the way of resting (active or passive), and question 13a is related to the appropriateness of engaging in physical activity such as dancing in relation to age and/or social status. However, all the remaining 22 questions were characterized by high reliability (ICC_3.1_ in the range of 0.78 to 0.97). So far, the reliability of the Polish version of the KCS questionnaire has not been assessed in people with chronic LBP. However, this questionnaire has already been used to assess the effectiveness of therapy in people with LBP [68]. In these studies, before the implementation of therapy, similar values of both domains and the overall kinesiophobia index were recorded. On the other hand, in studies on healthy women at perimenopausal age (48–58 years), a similar value of the biological domain was recorded, but higher values of the psychological domain and the overall kinesiophobia index were recorded [69].

A limitation of this study is the overrepresentation of women compared to men (73.8% vs. 26.5%). A similar percentage of women participating in studies related to the assessment of the level of kinesiophobia is also indicated by other authors [32,70,71,72]. The reasons for this state of affairs can be seen, on the one hand, in the fact that men use medical care less often than women [73]. Therefore, in future studies, a larger percentage of men should be included in order to assess the impact of sexual dimorphism on the repeatability of results. Another limitation that could potentially affect the obtained research results is the failure to take into account the past and current physical activity of people suffering from chronic nsLBP. Another limitation may be the arbitrarily adopted 4-week interval between studies. Further studies should determine the optimal time interval in studies aimed at assessing the reliability of questionnaires.

## 5. Conclusions

The Polish version of the KCS questionnaire is characterized by good internal consistency and reliability, both for the biological and psychological domains, as well as for the KCS questionnaire as a whole. The KCS questionnaire is a good psychometric tool useful for assessing the psychosocial determinants of kinesiophobia among people suffering from LBP. The measurement of the determinants of kinesiophobia may be an important determinant of the effectiveness of therapy in people suffering from nsLBP. The structure of the KCS questionnaire, which separately analyzes the biological and psychological determinants of kinesiophobic behaviors, also makes it a useful tool for individualizing the selection of rehabilitation measures for people with nsLBP.

## Figures and Tables

**Figure 1 diagnostics-15-01746-f001:**
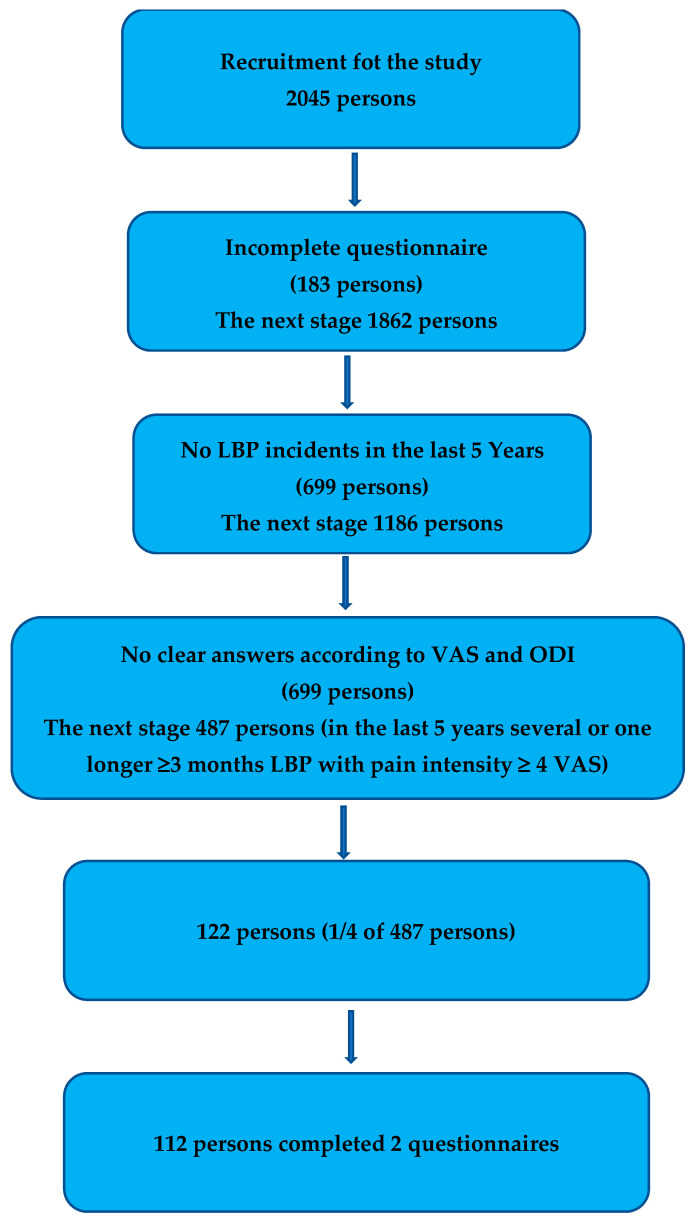
Stages of selecting respondents.

**Table 1 diagnostics-15-01746-t001:** Results of the internal consistency of the KCS questionnaire.

Parameters	α	95% CI	SEM
Biological Domain	0.8	0.7–0.8	6.2
Psychological Domain	0.8	0.8–0.8	8.4
Total score of Kinesiophobia Causes Scale	0.9	0.8–0.9	4.6

Legend: α—Cronbach’s alpha reliability coefficient; 95% CI—confidence interval; SEM—standard error of measurement.

**Table 2 diagnostics-15-01746-t002:** Questionnaire results in both studies and the intraclass correlation coefficient (ICC) for the KCS questionnaire.

Parameters	Completion of the Questionnaire		ICC_3.1_ (95% CI)	SEM
	First (Mean ± SD) (95% CI)	After 4 Weeks (Mean ± SD) (95% CI)		
Morphological	27.01 ± 22.1 22.85–31.16	28.46 ± 21.9 24.36–32.56	0.93 (0.90–0.95)	0.04
Individual Need for Stimulation	44.61 ± 17.6 41.30–47.91	45.16 ± 18.8 41.65–48.68	0.86 (0.80–0.90)	0.2
Energetic Substrates	28.72 ± 20.3 24.92–32.52	30.37 ± 21.0 26.44–34.31	0.89 (0.86–0.93)	0.1
Power of Biological Drives	44.64 ± 25.1 39.94–49.34	42.86 ± 23.1 38.54–47.17	0.88 (0.82–0.91)	0.3
Biological Domain	36.24 ± 14.5 33.53–38.95	36.71 ± 15.1 33.89–39.53	0.90 (0.86–0.93)	0.1
Self-Acceptance	34.71 ± 29.7 29.16–40.27	34.73 ± 26.9 29.68–39.77	0.91 (0.88–0.94)	0.4
Self-Assessment of Motor Predisposition	49.67 ± 23.5 45.26–54.07	49.22 ± 23.3 44.86–53.58	0.93 (0.91–0.95)	0.1
State of Mind	41.96 ± 24.3 37.42–46.50	41.29 ± 24.4 36.73–45.86	0.91 (0.87–0.94)	0.01
Susceptibility to Social Influence	64.29 ± 21.7 60.23–68.34	65.18 ± 21.3 61.19–69.17	0.91 (0.87–0.93)	0.1
Psychological Domain	47.65 ± 19.6 43.98–51.33	47.62 ± 18.7 44.13–51.11	0.94 (0.92–0.96)	0.2
Total Score of Kinesiophobia Causes Scale	41.96 ± 15.4 39.08–44.84	42.15 ± 15.3 39.28–45.02	0.93 (0.91–0.95)	0.01

95% CI—95% confidence interval; SEM—standard error of measurement.

## Data Availability

Restrictions apply to the datasets. The datasets presented in this article are not readily available because the data are part of an ongoing study.

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
