# Peer review of "Reliability of the Polish Version of the Kinesiophobia Causes Scale (KCS) Questionnaire in Assessing the Level of Fear of Movement Among People Suffering from Chronic Nonspecific Low Back Pain"

_diagnostics, 2025, doi:10.3390/diagnostics15141746_

Round 1

Reviewer 1 Report (New Reviewer)

Comments and Suggestions for Authors

Saulicz et al. evaluated the reliability and internal consistency of the Polish version of the Kinesiophobia Causes Scale (KCS) for assessing fear of movement in people with chronic nonspecific low back pain (cLBP). With a sample of 112 participants who completed the KCS twice over a 4-week interval, the authors found high test-retest reliability and satisfactory internal consistency for the biological and psychological domains, as well as for the total score. The study supports the KCS as a valid tool for understanding psychosocial contributors to kinesiophobia.

The study addresses an important gap in the validation of psychometric tools for assessing movement-related fear in people with chronic low back pain and provides a culturally adapted version of the KCS for Polish populations. However, some issues should be addressed:

Keywords: Please consider including only keywords available among PubMed MeSH Terms.

Materials and methods

Selection criteria are clearly outlined. However, two critical clarifications are needed:

Was the participants’ baseline or past physical activity measured or considered as a confounding factor? This is highly relevant in studies of movement fear and pain.

The choice of a 4-week interval for the test-retest would benefit from a justification in this section.

Results

The results section is rich in data, but it can be a bit difficult to read. Tables presenting data for 24 individual questions make the section dense and more complex to interpret. Authors should consider relocating detailed item-by-item statistics to Supplementary Materials, keeping key metrics in the main text for clarity.

Discussion

The authors successfully anchor their findings in the biopsychosocial model. Nonetheless, there is insufficient comparison with alternative validated scales. A comparative interpretation would strengthen the rationale for the KCS.

In addition to the current psychometric discussion, the authors should acknowledge that not all chronic low back pain is homogeneous. For example, Zaina F, Marchese R et al. (J Clin Med. 2023) showed that people with scoliosis may present distinct pain localizations and neurogenic patterns. This opens a meaningful line of inquiry as to whether different spinal pathoanatomical structures affect kinesiophobia profiles and whether the KCS is sensitive to such subgroups.

Conclusions

This section aligns with the results, but should also identify clinical implications and suggest future research directions, such as implementing the KCS as part of a rehabilitation outcome monitoring strategy, exploring its responsiveness to change post-intervention, or other.

Author Response

Reviewer 2 Report (New Reviewer)

Comments and Suggestions for Authors

AUTHOR LIST AND AFFILIATIONS
• The information on the corresponding author is currently incomplete. At least one author should be designated as the corresponding author.
ABSTRACT
• The abstract should be revised and structured according to the journal's guidelines. According to the Instructions for Authors of Diagnostics Journal, it states: "Systematic reviews and original research articles should have a structured abstract of around 250 words and contain the following headings: Background/Objectives, Methods, Results, and Conclusions. Background/Objectives: A few sentences to place the question addressed in a broader context and highlight the purpose of the study. Methods: A brief description of the main methods or treatments applied. This can include any relevant preregistration or specimen information. Results: A short summary of the article’s main findings. Conclusions: A final summarizing comment of the main conclusions or interpretations. The abstract should be an objective representation of the article; it must not contain results which are not presented and substantiated in the main text and should not exaggerate the main conclusions. Clinical trial abstracts should include items that the CONSORT group has identified as essential."
• Abbreviations should be defined at the first mention in the text for low back pain in line 17, chronic nonspecific low back pain in line 19-20 and chronic nonspecific low back pain (nsLBP) in line 20.
• It is also recommended to include the keyword kinesiophobia as it is relevant to the topic of the study.
INTRODUCTION
• Ensure consistent terminology throughout the manuscript – it is recommended to standardise the use of 'non-specific' or 'non specific' or 'nonspecific' from lines 4, 20 and 38.
• In the introduction, it would be beneficial to add a clearly formulated hypothesis based on the stated aim of the study.
MATERIALS AND METHODS
• The subsection in the Materials and Methods, in line 84 - 89 that refers to ethical considerations, could be included in the Ethical Considerations section instead, or included only in the Institutional Review Board statement and Informed Consent Statement.
• The section describing the study design and participants lacks important information about the time period in which the study was conducted and the type of study (e.g. cross-sectional, longitudinal, etc.). In addition, the statistical values in this section should be corrected – instead of “(mean: 49.8; SD = 14.9)",” the recommended format is “(49.8 ± 14.9) years” in line 148.
• Furthermore, it is not clear from the manuscript why the questionnaires were completed again after four weeks. It would be helpful to justify this decision. I also suggest linking this point to the discussion in lines 250–258.
• In the section on statistical methods, abbreviations such as the intraclass correlation coefficient and the standard error of measurement should be inserted in brackets at the first mention (ICC, SEM), line 153 and 156.
RESULTS
• In the Results section, it is recommended to revise the title of Table 1 to better reflect the content, especially with regard to the assessment of the internal consistency of the questionnaire.
• In the legend of Table 2, all table legends need to be explained and the statistical format (e.g. mean ± SD) standardised.
• Table 3 also requires revision in terms of the presentation of data (mean ± SD) and it should be clarified what the labels “a”, “b” and “c” mean in questions 8 and 13 of the biological and psychological domains respectively – this is currently not clear from the table.
• In addition, Table 3 is not referenced anywhere in the manuscript, except for a brief mention in the discussion, which should be corrected.
DISCUSSION
• In the Discussion section, it is recommended to strengthen the link between the results obtained and current studies. Likewise, most of the discussion is a repetition of what has already been mentioned.
• It is particularly important to include in the limitations the information that the time interval between the tests could be up to three months, as mentioned in the manuscript.
• It is unclear whether the section of the discussion in lines 264–271 refers specifically to Table 3. If so, please make this connection clearer in the text to make it easier for the reader to understand.
REFERENCES
• References are not written according to Instructions for authors.

Round 2

Reviewer 1 Report (New Reviewer)

Comments and Suggestions for Authors

The authors adequately addressed the issues raised. No further comments.

Author Response

We would like to thank the reviewer for all their feedback. 

Reviewer 2 Report (New Reviewer)

Comments and Suggestions for Authors

Dear authors, the corrected manuscript successfully addressed the comments, but not all the corrections according to my comments have been made. However, there are still a few minor but important issues that need to be resolved before the manuscript can be accepted:

  1. The hypothesis is still not clearly stated in the Introduction. I suggest explicitly formulating the hypothesis to help readers better understand the aim of the study and the expected outcomes.

  2. Numerical value is incomplete (line 269: "ICC = 0.70 and 72" should write "ICC = 0.70 and 0.72").

Please revise the manuscript accordingly and submit a final version for consideration. 

Author Response

We would like to thank the reviewer for all the feedback.

We have now addressed both of the points by stating the hypothesis and correcting the numerical value. 

If there are any further issues to be addressed, please let us know. 

This manuscript is a resubmission of an earlier submission. The following is a list of the peer review reports and author responses from that submission.

Round 1

Reviewer 1 Report

Comments and Suggestions for Authors

This is actually an important study, and it states that acute pain in patients is a situation where the patient restricts their movements in order not to feel pain or starts to move in a controlled manner in order not to provoke pain, and the pain turns into chronic pain in the process. The treatment is also based on this argument.

The basic starting point here is not very convincing because acute pain heals and passes depending on the cause and the patient forgets about it. Maybe it never happens again. Most patients continue their lives without any problems. This situation can be in a very limited patient group. There is a situation in this group where patients reject all kinds of treatment, including algological procedure. After all, we cannot generalize this group, which has other psychological problems.

Reviewer 2 Report

Comments and Suggestions for Authors

First and foremost, I would like to commend the authors for their dedicated effort and hard work in presenting this research. Their contribution to the field is valuable and deserving of recognition. However, there are several concerns and areas for improvement that I believe should be addressed.

-The introduction section is excessively long, and the second paragraph should be completely removed. Additionally, the authors should briefly mention low back pain and its outcomes, rather than going into too much detail. Generally, the introduction in a paper should consist of 4 or 5 paragraphs to maintain clarity and conciseness.

-Moreover, there is some repetition of information regarding the importance of addressing kinesiophobia. Streamlining this section could improve readability.

-Figure 1 needs to be revised. Additionally, it is incorrect that some words are written in uppercase while others are written in lowercase. Consistent capitalization should be maintained throughout the figure.

-Could the authors please provide a clearer explanation of the following sentence: 'On the horizontal VAS scale, the subjects marked the degree of pain intensity during the last pain incident'? Typically, when studies are designed, specific time periods are defined for reporting the VAS score. For example, the highest VAS score over the last two weeks could be reported instead of focusing only on the most recent pain episode.

-In the first paragraph of the discussion, please provide an overview of the main results of your study.

- The authors reported that : 90 women (80.4%) and 22 men (19.6%) were included. Could the authors address potential biases introduced by the overrepresentation of women in the sample size.

-Moreover, could the authors give information on the implications of lower reliability for BD 9 in practical application.

-Please rewrite the conclusion as a whole paragrapgh.

-Could the authors explain why a one-month interval was chosen for test-retest reliability rather than a longer or shorter period. A comparison with other studies would strengthen this choice

Reviewer 3 Report

Comments and Suggestions for Authors

The study tested the reliability of Polish version of KCS, and it is important to extend the diagnostic tools for assessing the level of Kinesiophobia. Some suggestions as follows:

1.        The numbers of two complete questionnaires are inconsistent in Figure 1 and the text “questionnaires were obtained from 112 people who completed them twice”. Please make sure and keep caution about it.

2.        The first two paragraphs of discussion should be placed in the introduction, because most of them are more like a background introduction, not related to results of this study and the purpose of testing reliability of KCS.

3.        Usually, both reliability and validity will be tested for a new questionnaire. This study only focused the reliability, if possible, it is better to also explain the validity in the discussion section.

4.        Please explain the external validity of Poland version of KCS. Whether this questionnaire is suitable for people suffering from nonspecific LBP of other countries.

5.        There are also some other tools to assess the level of Kinesiophobia as what the authors wrote in the introduction section. what differences between other tools and questionnaire used in this study are there?  What is the advantage of Polish version of KCS? Whether other tools can’t satisfy the assess purpose?

Reviewer 4 Report

Comments and Suggestions for Authors

the paper was written very well. This paper will be interesting for readers in this field. This version is acceptable to publish regard to my idea. Please only add clinical discussion of these findings in the discussion section. in addition add sample size calculation of your study in the method section.

Round 2

Reviewer 1 Report

Comments and Suggestions for Authors

I did not change my mind.

It sould be send a pain journal

Reviewer 2 Report

Comments and Suggestions for Authors

Thank you for your detailed responses to the reviewer comments. 

1- Thank you for providing detailed information regarding low back pain in your manuscript. However, I would like to emphasize that the primary focus of your study is the reliability of the Polish version of the Kinesiophobia Causes Scale (KCS) questionnaire in assessing the level of fear of movement. While LBP is undoubtedly an important medical and socio-economic issue, providing excessive background on this topic detracts from the primary purpose of your research.  The fact that readers have different expectations regarding the outlines of the background of the research problem does not allow freedom to deviate from certain rules when preparing a manuscript.

-Fried T, Foltz C, Lendner M, Vaccaro AR. How to Write an Effective Introduction. Clin Spine Surg. 2019 Apr;32(3):111-112. doi: 10.1097/BSD.0000000000000714. PMID: 30234565.

-Pasek J Writing the empirical social science paper: a guide for the perplexed. 2012. Available at: www.apa.org/education/undergrad/ empirical-social-science.pdf. Accessed August 1, 2017.

-Van Damme H. Steps to Writing an Effective Introduction. Acta Chir Belg. 2015 Jan-Feb;115:1. PMID: 26466390.

2- However, I have noted a discrepancy in the terminology used throughout the introduction. While the study focuses on nonspecific low back pain (NSLBP), the references cited predominantly pertain to general low back pain (LBP) without distinction. This inconsistency needs to be addressed to ensure clarity and accuracy. For instance, the statement:

“According to Breivik [21], 60% of LBP patients visited a physician 2 to 9 times, and 11% did so at least 10 times [21].”

This reference appears to discuss chronic pain in general and does not explicitly address nonspecific low back pain. Such generalizations may mislead readers regarding the scope of the study. I recommend carefully reviewing all references cited in the introduction and ensuring that they specifically relate to NSLBP, as this is the focus of your research.

3- Thank you for providing an explanation regarding the overrepresentation of women in your study. However, the rationale you provided is problematic for several reasons. First, attributing this overrepresentation to gendered differences in the use of medical care or susceptibility to anxiety and avoidance behaviors lacks specificity and does not align with the primary focus of your research, which is nonspecific low back pain (NSLBP). Furthermore, the reference to a study on the use of inpatient mental health services by Hispanic women is not directly relevant to your research population or topic and therefore does not support your argument.

I strongly recommend revisiting this explanation and basing it on data or literature specific to NSLBP or kinesiophobia, if available. If such data are unavailable, it would be more appropriate to acknowledge the imbalance in gender representation as a limitation of the study without making unsupported generalizations. Maintaining a focus on evidence directly related to NSLBP will strengthen the credibility of your discussion.

Comments on the Quality of English Language

Could be improved.